# Altered Extracellular Matrix as an Alternative Risk Factor for Epileptogenicity in Brain Tumors

**DOI:** 10.3390/biomedicines10102475

**Published:** 2022-10-03

**Authors:** Jody M. de Jong, Diede W. M. Broekaart, Anika Bongaarts, Angelika Mühlebner, James D. Mills, Erwin A. van Vliet, Eleonora Aronica

**Affiliations:** 1Amsterdam UMC Location University of Amsterdam, Department of (Neuro)Pathology, Amsterdam Neuroscience, 1105 AZ Amsterdam, The Netherlands; 2Department of (Neuro)Pathology, UMC Utrecht Brain Center, University Medical Center Utrecht, 3584 CG Utrecht, The Netherlands; 3UCL Queen Square Institute of Neurology, London WC1N 3BG, UK; 4Chalfont Centre for Epilepsy, Bucks SL9 0RJ, UK; 5Center for Neuroscience, Swammerdam Institute for Life Sciences, University of Amsterdam, 1098 XH Amsterdam, The Netherlands; 6Stichting Epilepsie Instellingen Nederland (SEIN), 2103 SW Heemstede, The Netherlands

**Keywords:** brain tumor, extracellular matrix, matrix metalloproteinases, epileptogenesis, low-grade gliomas, low-grade epilepsy-associated neuroepithelial tumors

## Abstract

Seizures are one of the most common symptoms of brain tumors. The incidence of seizures differs among brain tumor type, grade, location and size, but paediatric-type diffuse low-grade gliomas/glioneuronal tumors are often highly epileptogenic. The extracellular matrix (ECM) is known to play a role in epileptogenesis and tumorigenesis because it is involved in the (re)modelling of neuronal connections and cell-cell signaling. In this review, we discuss the epileptogenicity of brain tumors with a focus on tumor type, location, genetics and the role of the extracellular matrix. In addition to functional problems, epileptogenic tumors can lead to increased morbidity and mortality, stigmatization and life-long care. The health advantages can be major if the epileptogenic properties of brain tumors are better understood. Surgical resection is the most common treatment of epilepsy-associated tumors, but post-surgery seizure-freedom is not always achieved. Therefore, we also discuss potential novel therapies aiming to restore ECM function.

## 1. Epileptogenicity of Brain Tumors

Brain tumors can present with various symptoms, of which seizures are the most prominent [1]. Several factors are associated with the epileptogenicity of brain tumors, such as the type of tumor, World Health Organization (WHO) grade, anatomical location, size, time interval before resection and the age of the patient [2,3,4]. In particular, slow-growing glioneuronal tumors and paediatric-type diffuse low-grade gliomas are found to be highly epileptogenic [4,5]. Most likely, a slow growth rate gives enough time for reorganizations in the adjacent cortex or in other brain areas such as the hippocampus, leading to increased hypersensitivity and, ultimately, increasing the chance of epileptogenesis [4,6]. However, rapid-growing, adult-type diffuse gliomas, such as glioblastomas, can also cause seizures [7]. Because of the aggressiveness of these tumors, they are often removed relatively quickly after discovery and therefore epilepsy is short-lived [8]. In contrast, low-grade epilepsy-associated neuroepithelial tumors (LEATs) lead to long-lasting epilepsy [4,9,10]. LEAT is the umbrella term for tumors, such as glioneuronal tumors and paediatric-type diffuse low-grade gliomas, that cause early onset, drug-resistant epilepsy. Several tumors have been classified as LEATs, such as dysembryoplastic neuroepithelial tumors (DNET/DNT), gangliogliomas (GG), papillary glioneuronal tumors (PGNT), pilocytic astrocytomas (PA), pleomorphic xanthoastrocytomas (PXA), diffuse astrocytomas and oligodendrogliomas (Table 1) [3,4,9]. The most commonly found tumors that are characterized as LEATs are the glioneuronal tumors DNETs and GGs [9]. DNETs generally occur in the cortex of the temporal lobe (67.3%) and have the highest seizure prevalence and the lowest age of onset amongst all LEATs [11]. DNT is a glioneuronal neoplasm in the cerebral cortex of children or young adults, characterized by the occurrence of a pathognomonic glioneuronal element [3]. DNETs often cause intractable, focal epilepsy with loss of awareness, with an age of onset around 15 years of age and a seizure prevalence close to 100% [9,11,12,13]. Similar to the DNETs, GGs often occur in the temporal lobe (86%), but they can appear in any part of the brain [14,15,16,17]. GGs are well-differentiated, slow-growing glioneuronal neoplasms composed of a combination of neoplastic ganglion (large mature neuronal elements) and glial cells [3,18]. Their seizure prevalence lies around 80–90% and seizure onset is around 16–19 years of age [11]. The seizures caused by GGs are typically focal, characterized by loss of awareness, and patients generally do not respond to anti-epileptic drugs [9,12,13,19].

## 2. Known Risk Factors for Tumor Epileptogenesis

There are several factors that are known to influence the likelihood of seizure and/or epilepsy development in patients with tumors (Figure 1), including the location of the tumor. Skardelly et al., investigated a subset of gliomas and found that tumor location in the temporoinsular is a predictor for epileptogenesis [28]. In addition, 77% of the LEATs localized in the temporal lobe [4,29], indicating that the temporal location of the tumor indeed increases the risk of epilepsy. This corresponds with the fact that temporal lobe epilepsy with hippocampal sclerosis (TLE-HS) is the most common form of epilepsy in the adult population subjected to surgery [4,30]. Moreover, LEATs involving mesial temporal structures are generally associated with a more widespread epileptogenic network [4].

Furthermore, it has been suggested that the size of the tumor is a predictor of epileptogenesis [28]. In general, patients with smaller brain tumors have a higher risk of developing epilepsy. Although counterintuitive, this is probably due to the fact that larger tumors may lead to earlier noticeable symptoms compared to smaller tumors, which results in treatment or removal of the tumors before epilepsy develops. A study including 13 different types of brain tumors, including metastases, revealed that patients with tumors smaller than 64 cm^3^ have an increased risk of developing seizures when compared to patients with bigger brain tumors, independently of tumor type [28]. This is in contrast to the studies by Pallud et al., and Lee et al., who did not find a significant association between tumor volume and epileptic seizures [5,31,32]. However, taking into account the confounding effects of other risk factors, smaller tumor size is indeed associated with increased seizure risk in glioma patients [28], indicating the importance of anti-seizure drugs prophylaxis not only in patients with large tumors, but already at an early stage of tumor growth.

Age and sex are also risk factors for tumor-related epileptogenesis. Subsequently, males and patients younger than 60 years have an increased risk of developing seizures [28,32]. LEATs are characterized by early-onset epilepsy as more than 80% of tumors present with seizure onset before the age of 15 years [4,29]. Furthermore, a slight male predominance is also seen in GGs [33] and DNETs [34,35].

Several genetic predictors of epileptogenesis are found in large cohort studies [11,35,36,37,38,39,40,41,42,43,44]. A complete deletion of the short arm of chromosome 1 and the long arm of chromosome 19 (1p/19q co-deletion) has been shown to increase the risk of epileptogenesis in paediatric-type diffuse low-grade gliomas [35,36]. Essentially, all 1p/19q co-deleted oligodendrogliomas also have a mutation in isocitrate dehydrogenase (*IDH*) *1* or *IDH2* [37,38]. *IDH1* and *IDH2* encode for two IDH enzymes involved in cytoplasmic (IDH1) and mitochondrial (IDH2) conversion of alpha-ketoglutarate to D-2-hydroxyglutarate (d-2-HG). Glioma cells secrete d-2-HG, which is structurally similar to glutamate, into the extracellular space [45]. In combination with the altered uptake and release of glutamate itself by glioma cells (reviewed in detail by others [46,47,48,49]), this can lead to an imbalance between inhibition and excitation and therefore cause seizures [50]. Furthermore, the IDH enzymes seem to play a crucial role in the response to oxidative and energetic stress and in cellular protection [39], processes which might be involved in tumor-related epileptogenesis. These mutations, however, are not decisive for the development of epilepsy, as *IDH1* and *IDH2* mutations do increase the risk of developing seizures in gliomas [51,52,53,54], while LEATs typically lack *IDH1* or *IDH2* mutations and 1p/19q co-deletions [11,40,41,42]. Moreover, recent data suggest that the mTOR pathway hyperactivation by d-2-HG is a potential mechanism of epileptogenesis in patients with IDH-mutated gliomas [3,55,56].

In LEATs, molecular alterations that can be linked to epileptogenesis are found in the Rat sarcoma/mitogen-activated protein kinase/extracellular signal-regulated kinase (RAS/MAPK/ERK) pathway and the phosphoinositide 3-kinase/protein kinase B/mammalian target of rapamycin (PI3K/AKT/mTOR) pathway. In particular, fibroblast growth factor receptor 1 (*FGFR1*) alterations and proto-oncogene B-Raf (*BRAF*) *V600E* (c.1799T > A) mutations are regularly found in LEATs [8,60]. Activation of the FGFR pathway initiates activation of the RAS/MAPK and the PI3K/AKT/mTOR pathways [58,59]. The mTOR pathway is involved in a lot of cellular processes that contribute to epileptogenesis, including cell growth, proliferation, cell survival and synaptic strength (reviewed by [61]). Several studies show a correlation between alterations in these pathways and epileptogenesis, indicating an important role for the mTOR pathway in epileptogenesis and its potential as a therapeutic target [60,62,63,64,65,66,67,68,69,70,71].

An emerging hypothesis is that epileptic seizures associated with gliomas may not be triggered by innate tumor properties such as tumor growth rate, molecular correlates and histopathological properties [32]. This was confirmed by meta-analyses in paediatric-type diffuse low-grade gliomas, meningiomas and a cohort of mixed brain tumors [32,72]. In search for additional indicators of epileptogenicity in tumors, we will here discuss the role of extracellular factors that, besides the intrinsic properties of the tumors, could also contribute to epileptogenesis.

## 3. Changes in Extracellular Matrix in Tumors and Epilepsy

Due to the extent of cellular reorganization that occurs during tumorigenesis and epileptogenesis, it is thought that changes in the brain extracellular matrix (ECM) are involved in the reorganization of epileptogenic tissue [73]. In healthy adults, around 20% of the total brain volume consists of ECM [74,75]. The ECM surrounds brain cells and is a dynamic microenvironment of the nervous system [75]. ECM molecules can interact with different cell-surface receptors affecting the communication between cells and a plethora of processes such as proliferation, migration, morphological and biochemical differentiation, synaptogenesis and synaptic activity [76,77,78]. Furthermore, molecules of the ECM can also interact with ion channels and neuromodulators, leading to regulation of synaptic transmission and, in pathological settings, to reduced firing thresholds [79]. This is clearly shown in several rodent models of seizures and epilepsy, where changes in the ECM composition led to increased excitability and occurrence of seizures [80]. As the extracellular proteins that make up the ECM are produced by the surrounding cells, changes in cellular activity and homeostasis might have profound effects on the cellular environment. Furthermore, glioma cells are known to secrete exosomes that interact with the peritumoral environment, promoting proliferation, angiogenesis and immunosuppression [81]. Recently, it was found that these glioma-secreted exosomes also modify the electrical properties and cause de-synchronization of surrounding neurons [82]. Considering the massive cellular alterations found in cerebral tumors including LEATs, it is not strange to imagine profound changes in extracellular proteins that accompany the pathological intracellular protein synthesis. However, studies concerning ECM changes in epilepsy-associated tumors are limited and, of the few, most focus on GGs. Taking the evidence from the field of epilepsy, we here seek to unravel the potential contribution of the ECM on the comorbid occurrence of seizures and cerebral tumors.

### 3.1. The Extracellular Matrix

There are various components of the ECM, which are classified according to their location in the brain: the basement membrane, interstitial matrix and perineuronal nets (PNNs) (Figure 2). The basement membrane is located between the cells of the neurovascular unit, mainly produced by endothelial cells, pericytes and perivascular astrocytes. It is mostly composed of collagen, laminin-entactin complex, fibronectin, dystroglycan and perlican [83,84,85]. The interstitial matrix is localized between neurons and glia and is mostly composed of hyaluronic acid, proteoglycans, tenascins, link proteins and a low amount of fibronectins and laminins [86]. PNNs are dense structures surrounding neuronal cell bodies and proximal dendrites and provide a physical scaffold for neuritic processes as well as a controlled microenvironment to other signaling proteins [87,88,89]. The most common components of PNNs include the chondroitin sulfate proteoglycans (CSPG), such as the lecticans neurocan, aggrecan, brevican and versican that attach to the abundantly present glycosaminoglycan hyaluronan forming the base and backbone of PNNs. Other components that make the structure rigid are tenascins (C and R), hyaluronan and proteoglycan link proteins (HAPLNs) [90,91,92]. The precise PNN composition can vary between cortical areas, and in vivo and ex vivo studies have shown that several components (such as link proteins and aggrecan) are more essential for PNN formation, while others (brevican, versican) appear to be less indispensable [93,94]. ECM proteins that make up the PNNs are produced and secreted by the neurons it surrounds as well as by astrocytes [95]. Interestingly, the expression depends on neuronal activity, as sensory deprivation can suppress PNN formation [96,97,98,99].

#### 3.1.1. Changes in ECM Proteins in Brain Tumors

GGs have a higher expression of ECM-related genes; the expression of laminins, collagens, thrombospondins, CD antigens, TIMPs and various integrins is higher in GGs than in control tissues [101]. Jaffey et al., showed a deposition of the ECM molecules laminin and collagen IV in GGs, but not in other gliomas [102]. Furthermore, the deposition of laminin and collagen IV is correlated to perivascular inflammation, leading to the hypothesis that these ECM molecules contribute to the typical phenotype that is seen in GGs [102]. Following the implantation of patient-derived glioblastoma multiforma (GBM) xenografts into mice, less PNN-surrounded cells were observed, a sign of disintegration of PNNs in peritumoral regions [103]. Furthermore, overexpression of various ECM components is observed in gliomas, including hyaluronic acid, brevican, tenascin-C, fibronectin and thrombospondin as well as specific integrins and other receptors interacting with ECM components, which lead to adhesion and migration of glioma cells [104,105]. Overall, this indicates a disturbed ECM in various brain tumors.

#### 3.1.2. Changes in ECM Proteins in Epilepsy

From studies in rats and humans, it is clear that the expression of various ECM proteins, such as tenascin-C, tenascin-R, neurocan and phosphacan is higher in the brain after seizures (reviewed in detail by [106,107]). Furthermore, gene expression of various ECM components changes throughout epileptogenesis in rat brain tissue [108,109], including thrombospondins, CD antigens, cathepsins, MMPs, and TIMPs. Induction of convulsions by systemic kainate injections in rodents resulted in changes of two major CSPGs in the hippocampus [110]. Specifically, a higher expression of neurocan was apparent 24 h after convulsions occurred, which accords with a decreased expression of phosphacan. Interestingly, the number of phosphacan-positive PNNs decreased after spontaneous seizures occurred. This pattern remained apparent in half of the animals two months after the induced convulsions [110]. Furthermore, aggrecan expression is dramatically reduced in PNNs for up to 2 months after status epilepticus in rats, preceded by a reduction in the expression of hyaluronan and proteoglycan link protein 1 [106,111]. In resected brain tissue of patients with temporal lobe epilepsy, alterations of several ECM components are evident, including glycosaminoglycans (GAGs) [112], and chondroitin sulfates [113,114]. This is confirmed by studies using rodent models of epilepsy [111,115,116].

#### 3.1.3. How CAN ECM Changes Contribute to Epileptogenesis?

The disruption of the ECM in the epileptogenic brain might lead to a less stable and more permissive environment in the brain, increasing possibilities for neuronal reorganization during epileptogenesis [106]. Indeed, depletion of ECM/PNN components has been shown to cause or facilitate seizures. Young et al., showed that hyaluronidase injections (leading to degradation of hyaluronan) cause the generation of spontaneous seizures in cats [111,117]. In addition, more recently, it has been shown that hyaluronidase-treated hippocampal neurons have altered surface mobility of AMPA receptors, thereby changing synaptic short-term plasticity [118]. Brevican also appears to modulate neuronal plasticity by regulating potassium channels as well as AMPA receptors [119]. In a rodent model of epilepsy, degradation of PNNs using chondroitinase ABC resulted in a lower PTZ-induced seizure threshold [111].

Important in considering the epileptogenicity of ECM alterations is the location of the respective ECM structures. In the hippocampus, which is often considered the generator of TLE (reviewed in Huberfeld et al. [120]), PNNs are mostly found around parvalbumin (PV)-expressing fast-spiking GABAergic inhibitory interneurons [121,122] as well as some excitatory neurons in the CA2 region of the hippocampus [123]. The local ECM structures reduce the plasticity of neurons by creating a physical barrier against the formation of new synapses [124].

Additionally, it was recently shown that the PNNs have another, previously unrecognized function due to its close relationship with the cell membrane. By increasing the membrane’s thickness, PNNs can have an insulating effect. Changes in proximity of the PNNs can therefore results in a change of membrane capacitance [103]. Finally, reduction of these ECM structures affects PV+ interneuron homeostasis. Indeed, a reduction of more than 30% in fast spiking interneurons was observed in peritumoral brain material [125]. Together, this indicates that the reduction of PNNs around the PV+ interneurons might make these cells more vulnerable for inhibitory inputs, making the area more excitable [126], which could also take place around tumors.

Interestingly, the abnormal neuronal activity caused by epileptic seizures can also lead to the reduction of PNNs in the cortex and hippocampus [106,110,115], potentially due to the activity-dependent secretion of metalloproteinases able to degrade CSPGs and other ECM components [127,128]. These are enzymatic factors that are observed in several brain tumors as well, indicating that a potential positive feedback loop might be involved in epileptogenesis in tumors.

In the following paragraphs, we will focus on specific ECM components that could play an important role in the development of epilepsy in brain tumors: matrix metalloproteinases, urokinase-type plasminogen activator receptor, leucine-rich glioma inactivated 1, and glypicans. We will provide some background for each of the components, summarize the findings in brain tumors as well as the epileptogenic brain, and explain how these specific ECM components may contribute to epilepsy in brain tumors.

### 3.2. Matrix Metalloproteinases

The ECM can be regulated and controlled by MMPs and a disintegrin and metalloprotease with thrombospondin motifs (ADAMTs) [129]. MMPs are calcium-dependent zinc-containing endopeptidases that are expressed with a pro-peptide that needs to be removed for activation [130]. ADAMTs are multi-domain matrix-associated zinc metalloendopeptidases that are structurally related to MMPs [129]. MMPs and ADAMTs are involved in the degradation of the ECM [131], tissue morphogenesis [132], neuroinflammation by the release of apoptosis ligands, activation of pro-inflammatory markers and chemokine inactivation [133,134,135]. Currently, 24 homologues have been identified, which can be divided into six sub-groups: collagenases, gelatinases, stromelysins, matrilysins, membrane-type metalloproteinases and others (reviewed in Rempe et al. [136]).

#### 3.2.1. Changes in Matrix Metalloproteinases in Brain Tumors

MMPs are known to play an important role in tumor pathology [33,137] as the MMP proteolytic system is involved in cell proliferation and other processes of metastasis, such as intravasation and extravasation of tumor cells [33,137], migration of tumor cells within the brain [138] and adhesion of metastatic cells to the ECM [137]. The MMPs that are known to be involved in paediatric-type diffuse low-grade gliomas brain tumors are mainly MMP2 and MMP7. MMP2 is higher-expressed in diffuse astrocytomas than in normal brain tissue, but lower than in adult-type diffuse gliomas [139]. Yu et al., show in a murine astrocytoma cell line that MMP2 expression is correlated with cell invasion and negatively correlated with survival and proliferation [140]. Additionally, serum levels of MMP7 are higher in patients with low-grade astrocytomas compared to healthy controls and it are also higher than in patients with high grade brain tumors [141]. Although little is known about the effect of higher MMP7 expression in astrocytomas, a correlation between MMP7 expression and tumor severity in thymic and esophageal tumors is shown [142,143].

#### 3.2.2. Changes in Matrix Metalloproteinases in Epilepsy

Because of their role in remodeling the ECM, MMPs are known to also play a role in epileptogenesis. From several studies it is evident that MMP expression is increased in resected brain tissue from epilepsy patients and in various animal models of epilepsy [62,144,145,146,147,148,149,150]. MMP9 protein expression is higher in the serum of TLE-HS patients, as well as its enzymatic activity in the hippocampi of these patients as compared to controls [144]. Konopka et al., investigated surgically resected epileptogenic brain tissue from patients with focal cortical dysplasia and observed higher MMP9 expression along with, although to a lesser extent, higher expression of MMP1, MMP2, MMP8, MMP10 and MMP13 as compared to controls [151]. Furthermore, Li et al., found that MMP9 expression in the cerebral spinal fluid of patients with generalized tonic-clonic seizures was higher compared to controls [62,152]. Suenaga et al., suggested a relation between higher serum MMP9 levels and prolonged seizures in children with acute encephalopathy [153]. However, in several studies, lower MMP2 and MMP3 serum levels were reported in epilepsy patients as compared to control [146,154,155]. In a recent study, higher protein expression of MMP2, MMP3, MMP9 and MMP14 was shown in the hippocampi of patients who died after status epilepticus, as well as in resected brain tissue of TLE patients with hippocampal sclerosis TLE as compared to controls [156].

The use of animal models has elucidated the functional aspects of MMPs in epileptogenesis. Pijet et al., found that MMP9 expression was higher in rodent brains and blood after traumatic brain injury and that MMP9 overexpression increased the chances of developing spontaneous seizures after traumatic brain injury, indicating a contribution of MMP9 to the development of (post-traumatic) epilepsy [157]. A study with PTZ-kindled mice showed enhanced MMP9 activity and expression in the hippocampus, [148,158]. In another study using a rat model of kainic acid (KA)-induced seizures, higher MMP9 mRNA and protein expression was shown, followed by an increase in enzymatic activity, in the dentate gyrus of the hippocampus of treated animals compared to non-treated animals [149]. Furthermore, lower *Mmp9* expression in areas undergoing neuronal cell loss was shown in this study. This pattern of MMP9 expression suggests its involvement in the remodeling of dendritic architecture with possible effects on synaptic physiology [149]. Two studies using the electrically-induced SE rat model of TLE showed higher expression of *Mmp2*, *Mmp3*, *Mmp9* and *Mmp14* during epileptogenesis as compared to controls [156]. KA-induced seizures in rats led to higher MMP activity in hippocampal neurons in a neuronal activity-dependent manner [150]. Moreover, recombinant MMP9, but not MMP2, induces pyramidal cell death in cultures. In turn, KA-induced neuronal activity can intensify this neuronal death induced by MMP9, suggesting that MMP9 is involved in excitotoxic neuronal damage and the consequential neuroinflammatory processes [150]. In KA-evoked epilepsy and PTZ kindling-induced epilepsy rodent studies, it is shown that both the MMP9 protein expression and enzymatic activity become strongly increased upon seizures [147]. Furthermore, in this study it is shown that MMP9 knock-out decreases seizure-evoked pruning of dendritic spines and decreases abnormal synaptogenesis after mossy fiber sprouting in these TLE rat models, indicating a potential mechanistic basis for the effect of MMP9 on epileptogenesis. In a study by Takacs et al., a correlation is reported between an increased enzymatic activity of MMP9 and occurrence of epileptic activity that did not cause cell death. The increase in MMPs that is seen in epilepsy is thought to contribute to epileptogenesis [147]. As we previously described, MMP9 knockout-mice are less sensitive to chemically induced seizures and vice versa [147,148].

#### 3.2.3. How Can Changes in Matrix Metalloproteinases Lead to Epilepsy in Brain Tumors?

As discussed earlier, MMPs are known degraders of several ECM proteins [131]. This could lead to the disruption of the structural support that is given to neurons, including interneurons. Indeed, it has been shown that MMPs target and disrupt PNN structures influencing the surrounded microenvironment [159]. This causes changes in receptor and channel distribution and altered ion buffering, eventually leading to dysfunction of the neuron and less inhibitory capacity in the circuit [103]. Though there is no specific involvement of interneurons in LEATs, MMPs can directly influence AMPA and NMDA receptors [160,161], potentially altering neuronal excitability in or around the tumor. Furthermore, MMPs can target tight junction and cell adhesion molecules at the blood–brain barrier, leading to its disruption [36,162] which can contribute to epileptogenesis [163]. Besides the increase in migration and metastasis that is subsequent to this in high-grade tumors, disruption of the blood–brain barrier can promote brain inflammation [135]. This is accompanied by the cleavage and activation of inflammatory factors by MMPs. Inflammatory processes are known to be involved in epileptogenesis (reviewed in Pitkänen et al. [164]), suggesting that higher MMP expression facilitates epileptogenesis, also in tumor environments. Interestingly, occurrence of seizures can also cause higher MMP expression (reviewed in Chakraborti et al., and Yan et al. [165,166]), suggesting that higher MMP expression could also be a consequence of epileptogenesis, leading to a vicious circle. The exact role of MMPs in LEATs needs to be further investigated, but given their large role in epilepsy and during epileptogenesis on the one hand [62,144,146,147,148,149,150], and in tumor development on the other hand [33,137], MMPs and other metalloproteinases can provide interesting targets for investigation and future therapies for LEATs.

### 3.3. Urokinase-Type Plasminogen Activator

Urokinase-type plasminogen activator receptor (uPAR) is an extracellular receptor attached to the cell membrane and consists of three homologous domains that are linked by a short linker region [167]. uPAR can bind protease urokinase-type plasminogen activator (uPA or urokinase) as well as the ECM protein vitronectin [167,168]. By docking uPA to uPAR, a proteolytic cascade is initiated in which uPA cleaves plasminogen, creating the protease plasmin, that subsequently can cleave and activate MMPs [169]. Both plasmin and MMPs are involved in the degradation of ECM. Via this way, uPAR regulates several tissue-remodeling processes such as adhesion, proliferation, receptor internalization, apoptosis and migration [170,171,172]. Through both initiating the proteolytic cascade and the downstream signaling, uPAR can contribute to plasticity in epileptogenic tissue. uPAR also acts as a docking receptor for vitronectin, by which it can induce reorganization of the cytoskeleton and influence cell migration [173].

#### 3.3.1. Changes in Urokinase-Type Plasminogen Activator in Brain Tumors

Increased uPAR expression is generally associated with high-grade gliomas and is thought to play a role in metastases and migration [174]. However, also in low grade tumors such as GGs, uPAR expression is also found to be increased [67,68]. uPAR is able to interact with beta integrins and, by recruiting EGF receptors, can regulate integrin activity [175]. This process leads to activation of the MAPK/ERK pathway causing induction of proliferation. uPA positively stimulated the interaction between uPAR and integrins [176]. Interestingly, increased expression of uPA is also seen in GGs and other gliomas [67,68,174]. Through the downstream pathways that are initiated upon interaction with integrins and uPA, uPAR provides an important player in controlling cell proliferation, migration and adhesion [175]. In GGs, uPA and uPAR might have a seizure-related effect, as the increased expression is also seen in epileptogenesis and epilepsy [68,177].

#### 3.3.2. Changes in Urokinase-Type Plasminogen Activator in Epilepsy

The plasminogen system is involved in many processes such as neuronal development, inflammation and synaptic remodeling, which are important processes for epileptogenesis [171,178,179]. Indeed, the expression of uPAR is increased in several animal models and human epilepsy [68,144,162,180]. Previous research showed a higher expression of uPAR during epileptogenesis in patients with different epileptogenic pathologies, such as TLE-HS and tuberous sclerosis complex (TSC) [68]. Interestingly, uPAR knockout mice exhibit spontaneous epileptic seizures and are more susceptible to induced seizures [67,68,174,175,176,177,179,181,182].

#### 3.3.3. How Can Changes in Urokinase-Type Plasminogen Activator Lead to Epilepsy in Brain Tumors?

The increased seizure susceptibility that is observed in uPAR knock out models indicates that dysregulation of the uPAR and its downstream pathways can alter circuit excitability. Powell and colleagues observed impaired migration of inhibitory GABAergic neurons during development in uPAR knock out mice along with a substantial decrease in GABAergic interneurons in anterior cingulate and parietal cortices [179]. This was mostly driven by a virtually complete loss of parvalbumin positive interneurons [179,181]. It is therefore interesting that increased uPAR expression is found in epilepsy patients. Given the important role of interneurons in epileptogenesis, future studies should focus on entangling the effect of this increased expression in relationship to loss of interneurons and cell migration. So far, several uPAR-targeting therapeutic strategies have been explored in oncology, but none have been tested in clinical trials [183]. Going forward, inclusion of epilepsy-related brain tumors in such preclinical studies will provide a better insight in the role of uPAR in epileptogenesis in tumors.

### 3.4. Leucine-Rich Glioma Inactivated 1

The leucine-rich glioma inactivated 1 (LGI1) gene encodes for a glycoprotein that is highly expressed in the CNS, specifically in the hippocampus [184]. The LGI1 protein is secreted by neurons and binds to the cell surface where it interacts with several ECM-associated membrane-anchored proteins of the ADAMs family. LGI1 is composed of a C-terminal epitempin repeat domain and an N-terminal domain that is enriched in leucine (leucine-rich repeat domain).

#### 3.4.1. Changes in Leucine-Rich Glioma Inactivated 1 in Brain Tumors

Besides its anti-epileptogenic effects, LGI1 is also hypothesized to be a tumor suppressor gene. LGI1 is downregulated in paediatric-type diffuse low-grade gliomas and malignant gliomas, as well as in several tumors that can be classified as LEATs [185]. LGI1 impairs proliferation and survival and increases cell death and apoptosis in HeLa cells [186]. Moreover, LGI1 overexpression inhibits the expression of MMPs, specifically MMP1 and MMP3, in glioma cells through the MAPK/ERK pathway [187]. Decreased MMP expression in brain tumors leads to reduced proliferation, invasion and neovascularization [188,189]. LGI1 is also expected to be involved in suppression of cell migration, as increased expression of LGI1 in glioma cell lines reduced the proliferation and migration ability of these cells significantly [190]. Altogether, there are strong indications that LGI1 is a tumor suppressor gene, although some studies debate this. Krex et al., could not detect a mutation in the LGI1 gene in any of the 11 tested glioma cell lines [191] and it can be argued that evidence proving the tumor-suppressive role of LGI1 is limited (reviewed in [192]).

#### 3.4.2. Changes in Leucine-Rich Glioma Inactivator 1 in Epilepsy

A mutation in the LGI1 gene has been identified to play a role in epileptogenesis as 19 different LGI1 mutations have been described in 22 families with epilepsy [193,194,195,196,197]. It is proposed that a lack of LGI1 due to missense mutations disrupts synaptic protein connection and selectively reduces synaptic transmission in the hippocampus and thus serving as a determinant of brain excitation [198,199]. Furthermore, the loss of these proteins is known to cause spontaneous seizures in mice, indicating LGI1 could be a very important antiepileptogenic protein [31,198,200].

#### 3.4.3. How Can Changes in Leucine-Rich Glioma Inactivator 1 Lead to Epilepsy in Brain Tumors?

Little is known about the involvement of LGI1 in the epileptogenesis in LEATs. However, several studies have shown the association of LGI1 and membrane channels and synaptic firing. For example, Fukata et al., show that, after incubation with LGI1 alkaline-phosphatase, the ratio between evoked synaptic currents by AMPA versus NMDA receptors increases in hippocampal slides of mice brains [199]. Furthermore, in this study it is shown that the amplitude and frequency of AMPA-mediated postsynaptic currents are increased. *Lgi1* mutation results in impaired function of the LGI1 protein, leading to reduced AMPA receptor activity in rat hippocampal slices [199]. It is suggested that this deficiency might lead to an increase in excitatory signals compared to inhibitory signals and therefore increasing the change of epileptogenesis. However, LGI1 is more often expressed in tumor tissues of adult-type diffuse glioma patients with epilepsy than without epilepsy [201]. An explanation for this paradoxical finding is the presence of autoantibodies to LGI1 in the sera of these patients, which might have a net epileptogenic effect [201]. As it is known that the expression of LGI1 is lower in several LEATs and other low-grade gliomas and that lower expression of LGI1 is involved in epilepsy patients and mouse models, we argue that LGI1 is a promising potential therapeutic target in the treatment of epileptogenic brain tumors in general, and LEATs specifically, although more research is needed to specify its role and the role of LGI autoantibodies exactly.

### 3.5. Glypicans

Glypicans have been suggested as ECM components of interest as a few of them are identified as drivers of both tumorgenesis and epileptogenesis. Glypicans are part of the heparan sulfate proteoglycan families and have three key features: a protein core (60–70 kDa), heparan sulfate chains and a glycosylphosphatidylinositol linkage. The glycosylphosphatidylinositol links the glypicans to cell surfaces. There are six subtypes of glypicans called glypican-1 (GPC1) to GPC6, which differ in the primary sequence of their protein core and number of heparan sulfate chains [202]. They are involved in development and cell growth through regulations of WNT signaling [203]. They are mainly differentially expressed in liver and pediatric cancers but have also been found to be overexpressed in gliomas and are crucial for cancer cell growth and progression [204,205].

#### 3.5.1. Changes in Glypicans in Brain Tumors

In tumors, most of the GPCs have been found to be differentially expressed. Studies mainly showed a differential expression of GPCs in liver and pediatric cancers, but changes in expression have also been found in brain cancers, although no studies investigated LEATs specifically. GPC1 is linked to proliferation, angiogenesis and metastasis and is overexpressed in gliomas [206,207]. GPC2 is highly expressed in neuroblastomas but not detectable in other tissues, making it an ideal target for cancer therapy [208,209]. GPC3 works as a coreceptor for the tumor-regulating WNT protein [203] and can regulate proliferation of (tumor) cells by modulating Wnt, YAP and Hedgehog signalling [204,210,211]. Furthermore, GPC3 has been shown to be involved in cell growth and migration. Although GPC3 is mostly expressed in liver cancers and even serves as a diagnostic tool and potential treatment target in these tumors [210,212], it is also overexpressed in several LEATs such as oligodendrogliomas and astrocytomas [213].

#### 3.5.2. Changes in Glypicans Epilepsy

Of the glypican family, GPC3 and GPC4 have been implicated in the development and progression of epilepsy [214,215]. GPC3 was identified by Yu et al., as a driver of both tumorgenesis and epileptogenesis in a model of glioblastoma [215]. They observed that deletion of GPC3 resulted in a loss of the early onset hyperexcitability seen in control litter mates, while overexpression exacerbated the early onset of convulsive seizures [215].

Glypican 4 (GPC4) plays an important role in (excitatory) synapse formation and axon guidance, which are key pathological findings in patients with TLE [216]. GPC4 expression is elevated in neurons and astrocytes in the brains of epilepsy patients and in the neocortex and hippocampus of epileptic rats [214]. Ma et al., show that *Gpc4* promotes mossy fiber sprouting through the mTOR pathway and suppression of *Gpc4* reduces spontaneous recurrent seizures in a mouse SE model [203,204,206,207,208,209,210,211,212,213,217]

#### 3.5.3. How Can Changes in Glypicans Lead to Epilepsy in Brain Tumors?

Only very little is known about the epileptogenic effects of GPCs and their role in brain tumors and specifically LEATs, but the overexpression of glypican family members in both brain tumors and epilepsy brain tissue indicate the involvement of overlapping pathways. Indeed, the proposed mechanisms of action of GPC3 in tumorigenesis are also relevant in the development of epilepsy. Specifically, Wnt signalling, for which GPC3 acts as a regulator, is also known to regulate several processes, including neurogenesis and neuron loss, that are known to influence seizure development. Moreover, modulation of the Wnt signalling pathway can alter hippocampal sclerosis, a common hallmark of temporal lobe epilepsy [218,219]. GPC3 has also been shown to induce aberrant formation of excitatory and inhibitory synapsis, leading to changes in network excitability [215]. Additionally, the tissue specificity of GPC2 and the fact that its expression in increased in GNT patients with epilepsy, makes this an interesting target for future research to elucidate the exact role of GPCs in epileptogenesis in tumors.

## 4. Targeting the Extracellular Matrix as Novel Therapeutic Approach

LEATs are often drug-resistant and, consequently, the most frequently used therapy is surgical resection of the tumor [220,221,222,223]. Seizures are the first and foremost problem that patients with LEATs face and seizures do not always disappear after surgical resection of the tumor [224]. Therefore, there is a need for an additional or even a substitute treatment. There are several drugs in trial phases and already available that (indirectly) modulate the ECM and are proven to be antiepileptogenic. Here we discuss potential therapies that affect the ECM and could possibly be used for the treatment of epilepsy in epilepsy-associated tumors.

A potential therapeutic target during epileptogenesis is the PNN. Morwaski et al., show that administration of polyclonal antibodies against aggrecan can prevent the loss of PNNs in tenascin-R deficient mice [225]. So far, the direct effect of targeting PNNs on clinical symptoms has yet to be elucidated. More studies are needed to investigate the therapeutic potential of this antibody administration, but because of the ECM-restoring properties, it could be a potentially very effective treatment for patients with LEATs.

Another proposed therapy is altering the expression of ECM molecules that have an inhibiting effect on epileptogenesis such as aggrecan. Although, so far, the effects of already available and FDA-approved drugs have not been tested in neural cells, a few potential drugs have been identified in other cell types [226,227]. It has been shown that the L-type calcium channel blocker Verapamil increases the expression of aggrecan and collagens by increasing the gene expression of Frizzled-related protein (*FRZB*), an antagonist of the WNT-signalling pathway [226]. However, Verapamil is also a P-glycoprotein inhibitor and can aid in getting cytostatics past the blood–brain barrier [228,229]. Therefore, it is not a specific drug and treatment with Verapamil can have (unforeseen) side effects [230]. Verapamil as a possible treatment in drug-resistant epilepsy was reviewed by Nicita et al. [231]. A huge reduction or even cessation of epileptic seizures and status epilepticus is shown after treatment with Verapamil [232,233,234]. Furthermore, Lehmann et al., show that the anti-tuberculosis drug Isoniazid also increases aggrecan expression [227], secondary to its function as an antibiotic drug [235]. Although the number of studies using these drugs are limited and did not include neural cells, it might be very interesting to further investigate these already available drugs for the treatment of LEATs.

Based on recent experimental studies, the administration of MMP inhibitors could be considered [236,237]. Pollock et al., show that the broad-spectrum MMP inhibitor doxycycline prevents the re-organization of inhibitory innervation, PNN breakdown and seizure genesis [238]. However, so far, most clinical trials using MMP inhibitors as anticancer agents have failed due to technical problems, such as poor trial design, or MMP-inhibitor-related problems, such as metabolic instabilities and dose-limited toxicity and off-target effects (reviewed by Vandenbroucke et al. [239]). Initially, MMP inhibitors were designed as broad-spectrum inhibitors that would block MMP activity by binding to the zinc ion in the active sites of the MMPs. Minocycline is a broad-spectrum MMP inhibitor which is also used in epilepsy patients and is known to decrease seizures [240,241,242,243] but as most broad-spectrum MMP inhibitors, minocycline can lead to serious side effects such as vestibular problems and leukopenia [244,245,246]. Recently, IPR-179 (Act-03), a novel MMP2/MMP9-selective inhibitor, has been developed, which has antiseizure as well as antiepileptogenic effects and attenuates seizure-induced cognitive decline, without severe side effects, in two rodent models of epilepsy [156]. Therefore, this MMP inhibitor deserves further investigation in clinical trials.

Because of the dual role of LGI1 as both an anti-epileptogenic and tumor-suppressive protein that inhibits MMP expression, LGI1 could be an excellent therapeutic target for epileptic tumors in general and LEATs specifically [247].

Another possibility to regulate MMP expression is with microRNAs (miRNAs). miRNAs are small non-coding RNAs that can regulate gene expression by binding to the mRNA. Because of these gene-regulating properties, they are being used as a therapy, mostly in cancers (reviewed in Simonson et al. [248]). Several miRNAs are known to target MMP expression. For example, Korotkov et al., found higher mRNA and protein expression of *Mmp3* along with miRNA-155 expression in the hippocampus of TLE-HS patients and during epileptogenesis in a rat TLE model [249]. Furthermore, it is shown that inhibition of miRNA-155 attenuates *Mmp3* mRNA and protein overexpression in human astrocyte cultures [249]. Broekaart et al., found higher MMP mRNA and protein expression in TSC patients and showed that transfection of patient-derived TSC astrocyte cultures with miR-146a or miR-147b can decrease IL-1β-induced *Mmp3* expression [250]. Higher expression of miR-146a has been shown in both experimental and human epilepsy, including GGs as compared to control. Moreover, miR-146a mimic injection after epilepsy onset in a mice model of KA induced seizures reduces neuronal excitability and acute seizures [251]. Qin et al., found that miR-320d is lower-expressed in gliomas and that transfection with miR-320d can decrease MMP2 protein expression in glioma cell lines [252]. Additionally, Bongaarts et al., show that transfection with miR-320d decreases *MMP2* expression in fetal astrocyte cultures [253]. These studies show a great potential for miRNAs as regulators of the ECM and thereby as treatment for brain tumors. The potential of miRNAs as a treatment of epilepsy is nicely reviewed by Brennan et al. [254]. There are currently a few miRNAs being tested in clinical trials and their effects are long-lasting and without serious adverse effects [255,256,257]. However, delivery of the miRNAs to the brain remains a challenge since miRNA molecules cannot easily cross the BBB. Possibilities are intranasal delivery or attachment of the miRNAs to carrier molecules that can cross the BBB, but further research is necessary to determine the efficacy of these methods for this specific use.

So far, several drugs and therapies have been identified that target the ECM. More studies are needed to determine the (side) effects of these therapeutics in the human brain, but targeting the ECM and restoring the changes that have been observed in epilepsy and brain tumors could be very promising for future epilepsy-related brain tumor treatment.

## 5. Conclusions

Besides tumor type, location, and genetics, the ECM could play an important role in both tumorigenesis and epileptogenesis. Therefore, specific ECM components may also be considered as therapeutic targets, and investigating the role of the ECM in epilepsy-associated brain tumors remains an important goal to better understand the progression of this pathology and to develop novel therapeutic strategies.

## Figures and Tables

**Table 1 biomedicines-10-02475-t001:** Epilepsy-associated tumors and their characteristics.

Tumor Type	Most Frequent Tumor Location	Seizure Frequency	Age of Seizure Onset	Drug Resistance	Duration of Epilepsy ^#^	WHO-Grade
DNET [3,4,9,11,13]	Temporal	100%	15 years	Yes	Long-term	Grade 1
GG [3,4,9,13,19]	Temporal	80–90%	16–19 years	Yes	Long-term	Grade 1
PGNT [3,4,9,13,20]	Temporal/frontal	34–50%	26 years	Yes	Short-term	Grade 1
PA [3,4,21]	Temporal, cerebellum	Unknown	14 years	Yes	Long-term	Grade 1
PXA [3,4,22]	Temporal	75%	10–30 years	Yes	Long-term	Grade 2,3
Diffuse astrocytoma [3,4,23]	Temporal/frontal	60–85%	20–40 years	Yes	Long-term	Grade 1
Oligodendro-glioma [3,4,24]	Frontal, temporal	70–90%	40–60 years	Yes	Short-term	Grade 2,3
Cohort of astrocytomas, oligodendrogliomas and oligoastrocytomas [3,13,25]	Temporal	80%	20–40 years	Yes	Long-term	Grade 1–2
cohort of anaplastic astrocytoma, anaplastic oligodendroglioma or glioblastoma [3,13,26]	Frontal/fronto-temporal	25–60%	60 years	No	Short-term	Grade 3–4
Meningioma [3,27]	Supratentorial meninges	25–40%	40–60 years	No	Short-term	Grade 1–3

DNET, dysembryoplastic neuroepithelial tumor; GG, ganglioglioma; PGNT, papillary glioneuronal tumor; PA, pilocytic astrocytoma; PXA, pleomorphic xanthoastrocytoma. WHO, World Health Organization. ^#^ Duration of epilepsy: short-term < 2 years, long-term > 2 years.

**Figure 1 biomedicines-10-02475-f001:**
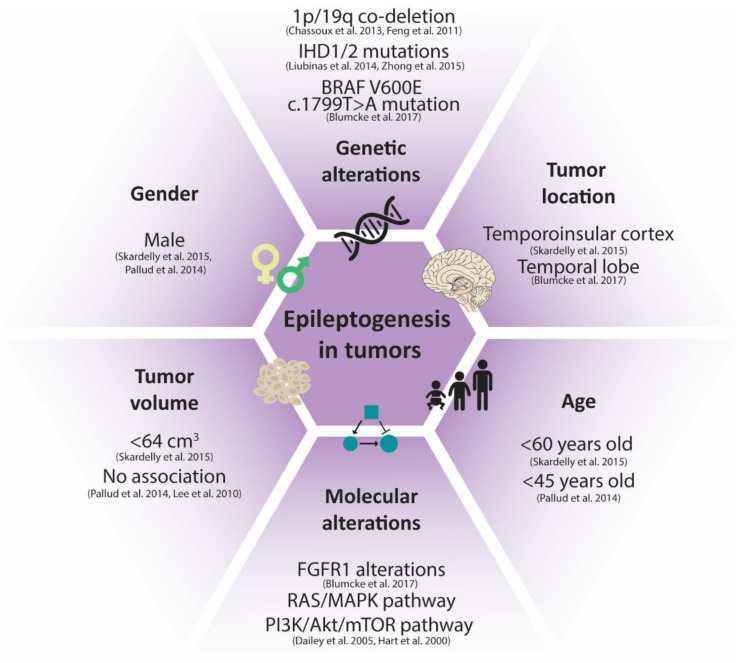
Known risk factors of epileptogenesis in tumors. Several clinical characteristics are linked to the occurrence of seizure and/or epilepsy development in patients with tumors, including the age and gender of the patients, as well as the tumor location and tumor volume. Additionally, several molecular and genetic alterations are associated with increased epileptogenesis risk in tumor patients. References clockwise starting from the top: [5,8,28,35,36,52,53,57,58,59].

**Figure 2 biomedicines-10-02475-f002:**
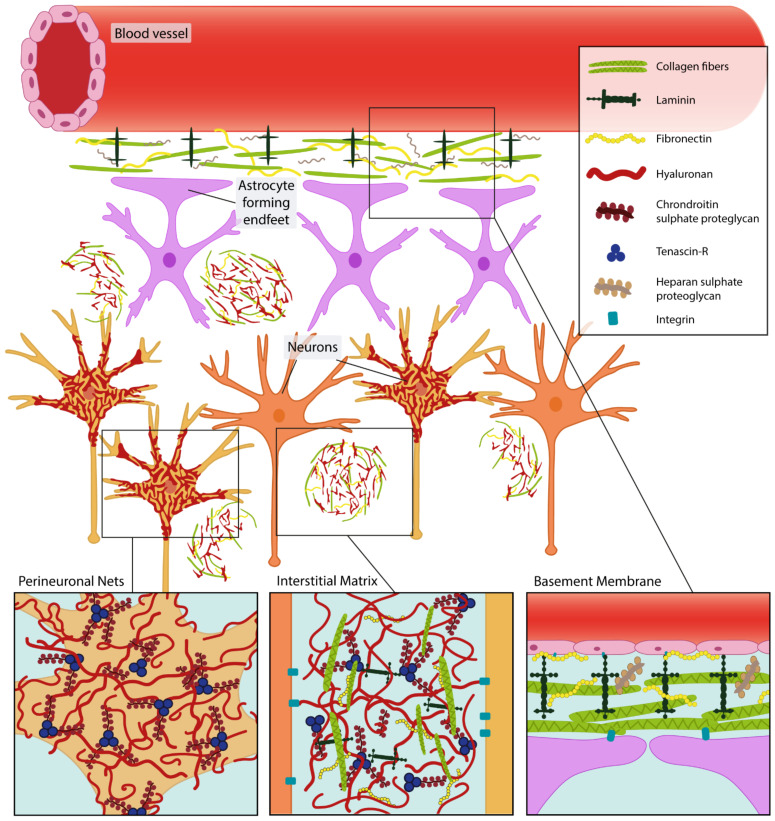
Schematic overview of the extracellular matrix. Specific extracellular matrix structures can be recognized in the brain. Perineuronal nets surround neurons and proximal dendrites (orange). Extracellular matrix proteins provide structural support and form microenvironments for cell interactions in the form of interstitial matrix. Furthermore, basement membranes are formed by extracellular proteins secreted by astrocytic endfeet (purple), pericytes (not depicted) and endothelial cells of blood vessels (pink) that together make up the blood–brain barrier. Inspired by Lau et al., 2013 [100].

## Data Availability

Not applicable.

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
