# Peer review of "Altered Extracellular Matrix as an Alternative Risk Factor for Epileptogenicity in Brain Tumors"

_biomedicines, 2022, doi:10.3390/biomedicines10102475_

Round 1

Reviewer 1 Report

The review “Altered Extracellular Matrix as an alternative Risk Factor for Epileptogenicity in Brain Tumors”, aimed to considered the ECM as an important structure in epileptogenic and tumorigenesis, as it involvement in modelling and remodeling neuronal connections and cell-cell signaling. Authors discussed tumor type, location, genetics and the ECM role.

       After long and effective discussion, considering the epileptogenicity of brain tumors, risk factors, ECM changes, transcriptome and novel therapeutic approach, they concluded that tumor type, location, and genetics, the ECM could play an important role in both tumorigenesis and epileptogenesis.

       The review is a very good topic for readers, especially considering the new factors involved in ECM and tumor's progress.

Author Response

We thank reviewer 1 for the very positive comments. Since there were no questions or issues raised by this reviewer, we assume we can proceed with the editorial process.

Reviewer 2 Report

The review by Jody M. de Jong et al, is focused on the action of the brain tumor growth on microenvironment with effect on extracellular matrix composition. The authors described how glioma tumor could create the potential risk to develop epilepsy in affected patients. They are characterised by epileptic discharges which severity depends by the localization, size, and genetic status of the tumor in the brain. Though, the pathogenesis of tumor-associated epilepsy (TAE) is not fully understood, the authors described in deep manner the alteration of specific components of the ECM that have key role in seizure onset. Nevertheless, the manuscript describes, in static manner, the communication between the cancer cells and ECM in the brain neglecting dynamic components such as oncometabolites and exosomes. Before publication the manuscript needs some improvements.

Minor revisions:

1.      The authors could include a short description on the role of exosome released by cancer cells. Glioma-derived exosomes interact with the environment to favour tumor, invasiveness, and immunosuppression and they have effect on neuron activity (R. Spelat et al 2022).

2.      The increase of glutamate release into the peritumoral environment should be discussed as well. The glioma cells have a reduced ability to uptake glutamate, as they do not express sodium (Na+)-dependent excitatory amino acid transporters.

3.      The authors should also discuss in the manuscript the potential role of the oncometabolites released in the brain by cancer cells. It is known that the IDH enzyme plays a crucial role in the response to oxidative stress which might be involved in tumor-related epileptogenesis.  IDH1 is expressed in the cytosol and peroxisomes that carries out the catalysation of isocitrate to alpha-ketoglutarate. The mutated IDH1 enzyme converts alpha-ketoglutarate into D-2-hydroxyglutarate (D2HG), which gets secreted by the mutant
glioma cells into the extracellular space. D2HG is structurally similar to glutamate leading to an imbalance in excitation vs inhibition within the CNS and could thus potentially lead to seizures.

4.      The Authors could include more recent literature related to LG-1 tumor suppressor: i) Dazzo et al 2018 “LGI1 tumor tissue expression and serum autoantibodies in patients with
primary malignant glioma”; ii) Padmaja Kunapuli, et al 2004.
